# MIXTURE OF PARROTS 🦜🦜🦜: EXPERTS IMPROVE MEMORIZATION MORE THAN REASONING

**Samy Jelassi**[*]
Harvard University

**Clara Mohri**
Harvard University

**David Brandfonbrener**
Harvard University
Kempner Institute

**Alex Gu**
MIT

**Nikhil Vyas**
Harvard University

**Nikhil Anand**
Harvard University
Kempner Institute

**David Alvarez-Melis**
Harvard University
Kempner Institute

**Yuanzhi Li**
Microsoft Research

**Sham M. Kakade**
Harvard University
Kempner Institute

**Eran Malach**
Harvard University
Kempner Institute

## ABSTRACT

The Mixture-of-Experts (MoE) architecture enables a significant increase in the total number of model parameters with minimal computational overhead. However, it is not clear what performance tradeoffs, if any, exist between MoEs and standard dense transformers. In this paper, we show that as we increase the number of experts (while fixing the number of active parameters), the memorization performance consistently increases while the reasoning capabilities saturate. We begin by analyzing the theoretical limitations of MoEs at reasoning. We prove that there exist graph problems that cannot be solved by any number of experts of a certain width; however, the same task can be easily solved by a dense model with a slightly larger width. On the other hand, we find that on memory-intensive tasks, MoEs can effectively leverage a small number of active parameters with a large number of experts to memorize the data. We empirically validate these findings on synthetic graph problems and memory-intensive closed book retrieval tasks. Lastly, we pre-train a series of MoEs and dense transformers and evaluate them on commonly used benchmarks in math and natural language. We find that increasing the number of experts helps solve knowledge-intensive tasks, but fails to yield the same benefits for reasoning tasks.

## 1 INTRODUCTION

The explosion in capabilities of large language models in recent years has largely been enabled by scaling their size, as measured by the number of parameters in the model. In the standard Transformer architecture, scaling the number of parameters entails a proportional increase in computational cost, e.g. doubling the number of parameters requires doubling the number of floating-point operations (FLOPs), making training and inference more computational intensive. Mixture-of-Experts (MoE) were introduced as a solution for this problem (Shazeer et al., 2017; Lepikhin et al., 2020; Fedus et al., 2022). MoEs replace the single MLP in each Transformer block with multiple MLPs (called experts), where each token is routed to a few experts based on a linear routing function. The number of parameters in the MoE layer therefore increases with the total number of experts, while the compute increases only with the number of "active" experts (i.e., the number of experts to which the token is routed to). This offers a promising option for scaling models: increase the number of experts instead of the model dimension or its depth. For this reason, MoEs have become very popular, and many frontier models today are based on the MoE architecture (Achiam et al., 2023; Databricks, 2023; Anil et al., 2023; Dai et al., 2024; Jiang et al., 2024; Yang et al., 2024).

---

[*]Correspondence to: Samy Jelassi <sjelassi@fas.harvard.edu>

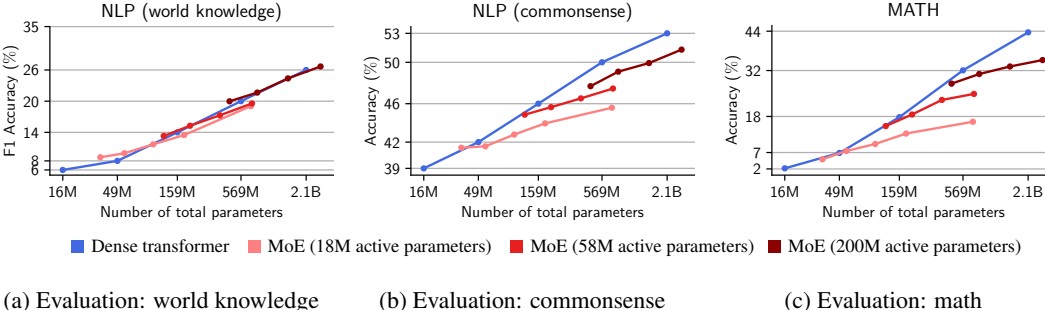

(a) Evaluation: world knowledge   (b) Evaluation: commonsense   (c) Evaluation: math

Figure 1: **(a) Evaluation: world knowledge.** We train a series of dense transformers and MoEs on 65B tokens from a corpus essentially made of Fineweb-edu, Cosmopedia and Wikipedia (see Section 5 for details). We then evaluate the models on several world knowledge benchmarks (e.g., TriviaQA (Joshi et al., 2017), Natural Questions (Kwiatkowski et al., 2019)) and report the average F1 accuracy. Surprisingly, at a fixed number of total parameters, MoEs with substantially fewer active parameters approximately match the performance of dense models. This highlights the importance of experts in tasks that require memorization. **(b) Evaluation: commonsense.** Here we evaluate the aforementioned pre-trained models on natural language commonsense benchmarks (e.g., HellaSwag (Zellers et al., 2019), WinoGrande (Sakaguchi et al., 2021)). On these reasoning tasks, we observe that MoEs perform worse than dense models and more significant benefits are obtained by increasing the number of active parameters. **(c) Evaluation: math.** Here we train a series of dense transformers and MoEs on 65B tokens from a corpus essentially made of Proof-Pile2 (Azerbayev et al., 2023) (see Section 5 for details). The results are consistent with the ones in (b): MoEs perform worse than dense models at equal number of total parameters.

In this work we study whether MoE indeed offers a "free-lunch", enabling gains in performance with no computational cost. Interestingly, we find that the benefit from MoEs greatly depends on the task at hand. We show that for reasoning-based tasks, such as graph problems and mathematical reasoning, MoEs offer limited performance gains, and increasing the number of experts cannot compete with scaling the dimension (width) of the model. On the other hand, for memory-intensive tasks, we show that scaling the number of experts is competitive with scaling standard "dense" MLPs.

To demonstrate these claims, we begin with a theoretical analysis of MoEs and dense models. We use communication-complexity lower bounds to show that a single-layer MoE requires a critical dimension to solve a simple graph connectivity problem, implying that MoEs offer no benefit for solving this problem and only consume unnecessary memory. On the other hand, we show that for a pure memorization task, where the model only needs to "remember" an arbitrary set of examples, scaling the number of experts is equivalent to scaling the number of parameters in dense transformers, implying a significant computational gain when fixing the number of active parameters (Section 3). We continue by experimentally validating these results, comparing MoEs against dense models on synthetic training data. We train these models on finding the shortest path in random graphs, where we show that MoE accuracy does not improve as we increase the number of experts, but that accuracy consistently increases with width for a dense transformer (Figure 4b). Following this, we train different models on the task of memorizing a large phone-book. We demonstrate that MoEs excel in memorization, matching the performance of dense transformers with the same number of total parameters but with substantially less computational cost (Figure 4a).

Finally, we train dense transformers and MoEs on real datasets of mathematical reasoning and natural language, and perform intensive benchmarking of these models on a wide variety of downstream tasks. For memory-intensive tasks, MoEs surprisingly have a great advantage, where increasing the number of experts can match the performance of large dense models (Figure 1a). However, we show that for tasks that rely on reasoning, scaling the number of experts cannot compete with increasing the model dimension (Figures 1b-1c). Moreover, MoEs exhibit some memorization behaviors when trained on math problems (Figure 5). Taken together, our results show that the gains from using MoEs depend greatly on the nature of the training data and downstream task, and that while MoEs can improve performance in certain cases, sometimes increasing the effective size (width) of the model is unavoidable.

## 2 RELATED WORK

**Mixture of Experts.** Mixture-of-Experts (MoE) date back to the work of Jacobs et al. (1991); Jordan & Jacobs (1994). Shazeer et al. (2017); Fedus et al. (2022) were the first to scale this idea to deep learning and obtain state-of-the-art models in machine translation. Since then, several works have improved their routing algorithms (Lepikhin et al., 2020; Lewis et al., 2021; Roller et al., 2021; Clark et al., 2022; Zhou et al., 2022; Antoniak et al., 2023; Zhong et al., 2024), have improved their downstream performance after finetuning (Du et al., 2022; Zoph et al., 2022) or made their training and inference more efficient (Rajbhandari et al., 2022; Gale et al., 2023; Pan et al., 2024; Tan et al., 2024). However, only a few papers have studied the science of MoEs and their comparison with dense transformers. Clark et al. (2022); Krajewski et al. (2024) establish scaling laws for MoEs. Chen et al. (2022) design a specific classification problem where a model with multiple experts provably outperforms one with only one expert. Shazeer et al. (2017); Lepikhin et al. (2020); Artetxe et al. (2021); Lewis et al. (2021); Fedus et al. (2022); Du et al. (2022) show that given a fixed FLOP budget, MoEs are always better. However, these papers claim that on a per parameter basis, MoEs always seem comparatively worse than dense models. In this paper, we temper this claim by showing that it depends on the *nature of the task* at hand: on reasoning tasks, we validate this claim but on memory-intensive tasks, equally-sized MoEs perform as well as dense transformers.

**Language models and memorization.** Large language models (LLMs) store a considerable amount of knowledge in their parameters (Petroni et al., 2019; Heinzerling & Inui, 2020). They memorize useful knowledge such as facts and commonsense (Zhao et al., 2024). Many works studied how memorization occurs in LLMs by developing tools to locate the knowledge in the model (Meng et al., 2022; Allen-Zhu & Li, 2023; Liu et al., 2024) or by tracking the training dynamics (Tirumala et al., 2022; Speicher et al., 2024). We draw inspiration from Allen-Zhu & Li (2023) and evaluate the memorization of our models by pre-training them on a mixture of datasets that includes Wikipedia, and at test time, evaluate them on world knowledge benchmarks, which are essentially question answering tasks on Wikipedia facts. With respect to theoretical findings, Kim et al. (2023); Mahdavi et al. (2023); Madden et al. (2024); Nichani et al. (2024) provide upper bounds on the number of parameters needed for dense transformers to perform memorization tasks under various conditions.

**Language models and reasoning.** In recent years, transformer-based language models have displayed remarkable effectiveness in solving a broad range of reasoning tasks. Specifically, the reasoning capabilities of transformers have been studied in the context of arithmetic problems (Jelassi et al., 2023; Cho et al., 2024; Hou et al., 2024; Zhou et al., 2024; McLeish et al., 2024; Lee et al., 2023), mathematical reasoning (Zhang et al., 2022; Imani et al., 2023; Wei et al., 2022) graph problems (Sanford et al., 2024; Fatemi et al., 2023; Jin et al., 2023; Wang et al., 2024) and code challenges (Shi et al., 2024; Zhu et al., 2024). Recently, state-of-the-art language models were used for solving complex math olympiad problems (DeepMind, 2024; NuminaMath, 2024; OpenAI, 2024). With respect to theoretical findings, various works study the reasoning capabilities of transformers, relating their expressive power to other complexity classes and formal languages (Weiss et al., 2021; Zhou et al., 2023; Strobl et al., 2024; Chen & Zou, 2024). Other works study how chain-of-thought can improve the reasoning capabilities of language models in terms of expressive power and learnability (Abbe et al., 2024; Merrill & Sabharwal, 2023; Malach, 2024).

## 3 THEORY: REPRESENTATIONAL CAPACITY

In this section, we analyze the capability of MoE transformers compared to standard (dense) models. We begin by studying a simple graph problem that requires scaling the hidden dimension of the transformer, showing that MoEs with small hidden dimension cannot solve this problem, regardless of the number of experts used. Then, we show that MoEs can effectively memorize random inputs, requiring significantly less computational resources (active parameters) compared to dense models.

### 3.1 SETTING

Consider a one-layer transformer $f \in \text{Transformer}_{m,H,1}^N$ which takes as input a sequence of length $N$ and has logarithmic bit-precision. $f$ embeds the input into dimension $m$ via the function $\phi$. $f$ has

$H \geq 1$ attention heads, whose outputs are combined via concatenation before we apply point-wise function $\psi$ [1].

Define the parameters as $Q_h, V_h, K_h \in \mathbb{R}^{m \times m}, \phi : \mathcal{X} \to \mathbb{R}^m, \psi : \mathbb{R}^m \to \mathbb{R}$. The output of $f$ is:

$$f(\boldsymbol{x}_1, \ldots, \boldsymbol{x}_N) = \psi\Big(\big[\operatorname{softmax}\big(\phi(x_N)^\top Q_h K_h^\top \phi(X)\big)\phi(X)V_h\big]_{h \in [H]}\Big).$$

$f$ is a *dense* transformer, if $\psi$ is an MLP, i.e. function of the form:

$$\psi(\boldsymbol{x}) = \boldsymbol{u}^\top \sigma(\boldsymbol{W}\boldsymbol{x} + \boldsymbol{b}), \text{ for } \boldsymbol{W} \in \mathbb{R}^{m' \times m}, b \in \mathbb{R}^{m'}, \boldsymbol{u} \in \mathbb{R}^{m'}$$

where $\sigma$ is the ReLU activation function. $f \in \text{Transformer}^N_{m,H,1,K}$ is a *sparse* (MoE) transformer with $K$ experts if $\psi$ is a function of the form:

$$\psi(\boldsymbol{x}) = \boldsymbol{u}_i^\top \sigma(\boldsymbol{W}_i\boldsymbol{x} + \boldsymbol{b}_i) \text{ for } i = \underset{j}{\operatorname{argmax}} \, \boldsymbol{r}_j^\top \boldsymbol{x}$$

where $\boldsymbol{W}_1, \ldots, \boldsymbol{W}_k \in \mathbb{R}^{m' \times m}, \boldsymbol{b}_1, \ldots, \boldsymbol{b}_k \in \mathbb{R}^{m'}, \boldsymbol{u}_1, \ldots, \boldsymbol{u}_k \in \mathbb{R}^{m'}$ are the parameters of each expert and $r_1, \ldots, r_k$ define the routing function (we use top-1 routing).

### 3.2 MoEs require a critical hidden size to solve graph reasoning tasks

In this section, we analyze the graph reasoning capabilities of dense and sparse transformers. We define the length-2 path problem on a graph, and use it as a means to understand other graph reasoning tasks such as graph connectivity, shortest path, and cycle detection.

**Definition 3.1** (Length-2 Path Problem). The input is a graph $G = (V, E)$. The source $s \in V$ and a destination $d \in V$ are fixed for all tasks as the 0 and $|V|$ vertex. The length-2 path problem asks whether there is a path of length 2 from $s$ to $d$.

Graph connectivity, shortest path, and cycle detection are all graph reasoning tasks which reduce to the length-2 path problem due to (Sanford et al., 2024) and Lemma D.2. We provide a lower-bound on the width required for a sparse transformer to solve the length-2 path problem, and an upper-bound on the width required for a dense transformer to solve the problem. Further, we show a separation between dense and sparse transformers with the same number of parameters: for a sufficiently large amount of experts in the sparse model, it cannot solve the same problem that a dense model can solve with the same amount of *total* parameters.

**Lower bound on width of depth-1 MoE for reasoning.** We begin by showing a lower-bound on the width for a depth-1 mixture of expert model for the length-2 path problem. This lower bound implies a lower bound for search and retrieval tasks such as graph connectivity, shortest path, and cycle detection.

**Theorem 3.2** (Length-2 path lower-bound on sparse transformers). *For some input sequence $G = (V, E)$, fix two disjoint subsets $A, B \subset [N - 1]$, and consider a single-layer transformer $f \in \text{Transformer}^N_{m,H,1,K}$ with $O(\log N)$-bit precision that solves length-2 path for any input $X$ where $X_A$ is a function of edges with the source $s$, $X_B$ is a function of edges with the destination $d$. Then, $f$ has width satisfying $mH = \Omega(|V|/\log N)$.*

The proof follows almost identically from the proof in (Sanford et al., 2024) for the class $\text{Transformer}^N_{m,H,1}$. The original proof does not place constraints on the function $\psi$ and is based on a communication-complexity argument. As such we may design $\psi$ so that it first routes and then chooses which expert to apply. We give a complete proof in Appendix D. As such, the result of (Sanford et al., 2024) can also be extended to the class $\text{Transformer}^N_{m,H,1,K}$.

**Upper bound on width of depth-1 dense transformer for reasoning.** In this section we give an upper bound for the width required for a dense model to solve the length-2 path problem.

---

[1] In multi-layer Transformers, each layer outputs a vector of size $m$. However, since our focus in this section will be on binary classification problems, we will let the transformer output a single scalar, and we interpret the output of the final token as the prediction for the classification task.

**Theorem 3.3** (Length-2 path width upper bound for transformer). *There exists a transformer of width* $|V|$, $H = 1$, *and* $O(\log N)$*-bit precision that solves length-2 path problem for any input.*

The proof relies on an encoding of the inputs where the output values only exceed a certain threshold when $u$ and $v$, the source and destination vertices, have edges with a common vertex. We defer the proof to Appendix D.

**Parameter-matched comparison of dense and sparse depth-1 transformers.** Using the lower-bound on width required for a sparse transformer (Theorem 3.2) and the upper-bound on width required for a dense transformer (Theorem 3.3), we compare dense and sparse transformers when they have the same number of total parameters. We find that when the number of experts exceeds $(\log N)^2$, the sparse model is unable to solve the same task as the dense model.

**Corollary 3.4.** *Consider a sparse transformer (with $K$ experts) and a dense transformer with the same number of parameters. There exists a number of experts $K$ so that the the sparse model is not able to solve the reasoning task, but the dense transformer solves the task.*

*Proof.* Suppose we have two depth-1 transformers, where one is a dense model and the other is a mixture of experts with $K$ experts. Let the width of the dense model be $m_d$, and the width of the sparse model be $m_s$. The number of parameters in the dense model is $O(m_d^2)$ and the number of parameters in the sparse model is $O(Km_s^2)$. In order to match the number of parameters, it must be the case that $m_s = \frac{m_d}{\sqrt{K}}$. Suppose we let $m_d = |V|$, as this is sufficient to solve the above problems. For any $K \geq \Omega\big((\log N)^2\big)$, the sparse model is not sufficiently wide to solve the problem. $\qquad\square$

### 3.3 MOES USE THEIR EXPERTS TO SOLVE MEMORY-INTENSIVE TASKS

In this section, we provide an upper-bound on the number of parameters necessary for a sparse transformer to solve memorization tasks, followed by a lower-bound on the number of parameters needed for a dense transformer to solve the same task. We use these results to compare the memorization capabilities of dense and sparse transformers with the same number of active parameters. We find that with enough experts, the sparse transformer is able to solve memorization tasks with less active parameters than the dense transformer. In both bounds we assume that transformer has logarithmic number of bits to encode each parameter.

We consider sequences $\{(X^i, y_i)\}_{i=1}^n$ where $X^i \in \mathbb{R}^{N \times m}$ are input sequences of length $N$ in dimension $m$ such that $X^i[j]$ is sampled from a Gaussian distribution $\mathcal{N}(0, I_m)$. We assume $y_1, \ldots, y_N \in \{\pm 1\}$ are arbitrary labels for the $n$ sequences. The objective is for a transformer to memorize these sequences, i.e. map each input $X^i$ to a label $y_i$. The classification is determined by the sign of the last token output.

**Upper-bound on MoE for memorization.** We begin by showing that, with high probability over the choice of the inputs, the MoE architecture can memorize (i.e., arbitrarily label the examples), with a small number of active parameters.

**Theorem 3.5.** *With probability at least* 0.99*, there exists a one-layer MoE transformer with $K$ experts, using $\tilde{O}\left(\frac{n}{K} + Km\right)$ active parameters and $\tilde{O}(n + Km)$ total parameters stored in $\tilde{O}(1)$ bits that, when applied to each sequence $X^i$, outputs at the last token a value whose sign matches $y_i$, i.e.,* $\operatorname{sign}(f(X_i)) = y_i$ *for all $i = 1, \ldots, n$.*[2]

Specifically, if we choose $K = \sqrt{n/m}$ we get that an MoE architecture can solve the memorization problem with $\tilde{O}(\sqrt{nm})$ parameters. To prove this, we show that for a particular routing function, the number of samples routed to each expert is approximately $n/K$. Then, we show that an expert with $\tilde{O}(n/mK)$ neurons can memorize a sample of size $O(n/K)$. We present the proof in Appendix D.2.

**Lower bound on memorization with dense Transformer.** Next, we give a lower-bound on the number of parameters for a dense transformer to perform memorization.

---

[2]We use $\tilde{O}$ and $\tilde{\Omega}$ to hide logarithmic factors.

**Theorem 3.6** (Lower bound for dense model). *Given the same task as above, a dense Transformer requires $\tilde{\Omega}(n)$ parameters to solve the memorization task.*

This bound follows from the fact that there are $2^n$ possible labels for any fixed set of $n$ inputs, and at most $2^{cW}$ functions with $W$ parameters and $c$ bit per parameters. The proof is in Appendix D.2.

**Separation between MoEs and Dense Models.** Observe that the previous results on memorization imply a separation between MoEs and dense models in terms of the number of active parameters. Namely, we show that an MoE with $\tilde{O}(\sqrt{nm})$ active parameters can memorize, while a dense model requires $\tilde{\Omega}(n)$ parameters. So, for $n \gg m$, MoEs are significantly more efficient. Comparing the number of total parameters, MoEs require $\tilde{O}(n + Km)$ parameters, so both MoE and dense models have linear dependence on $n$ in the total parameter count.

## 4 SYNTHETIC EXPERIMENTS

In the previous section, we proved that there exist graph connectivity problems that cannot be solved by any number of experts of a certain width but the same task can be solved by a dense model with a slightly larger width. Our goal in this section is to verify that our theoretical analysis bears out experimentally when training models from scratch on synthetic data, before moving on to study pre-trained models in Section 5. We mainly focus on two tasks: the *shortest path* problem (Figure 2), which we use as a synthetic task to represent reasoning problems, and the *phone-book* task (Figure 3), to measure the recall ability of our models. Our experiments in this section highlight that adding experts yields greater performance improvements on memorization tasks than reasoning tasks.

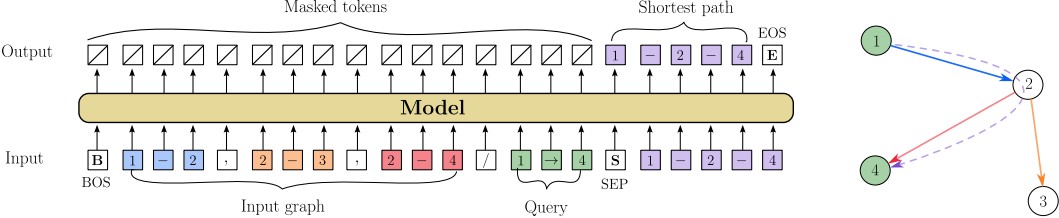

Figure 2: Illustration of the shortest path task. We feed the model with a sequence that lists all the edges in the input graph and ends with the query (in green) which asks the model to find a shortest path between two vertices (from vertex 1 to vertex 4 in the figure). The model then autoregressively returns the shortest path (in purple).

### 4.1 EXPERIMENTAL SETUP

**Architecture.** We opt for the Mistral (Jiang et al., 2023) and Mixtral (Jiang et al., 2024) architectures as the backbones of our Transformer and MoE models, respectively. The two architectures are identical and only differ by the addition of a gating module and multiple experts in Mixtral. For both model types, we fix the number of layers to $L = 12$. For the dense transformers, we vary model size by sweeping the width $d \in \{256, 512, 1024\}$. For MoEs, we sweep over widths $d \in \{256, 512\}$ and the number of experts $E \in \{8, 16, 32, 64\}$. To be consistent with our experiments in Section 5, we set the intermediate dimension in the FFN block to be equal to $d$ (and not $4d$). We use token-choice routing, do not apply any token dropping and each token is routed to the top-2 experts. Lastly, in both this section and Section 5, we report for each model the number of non-embedding parameters which we refer to as the total number of parameters.

**Shortest path task.** For a graph with $n$ vertices, our token space $\mathcal{V}$ is of size $n + 6$ with tokens encoding the vertices and some special tokens: $\mathcal{V} = \{1, \ldots, n, \langle \text{EDGE} \rangle, \langle \text{BOS} \rangle, \langle \text{EOS} \rangle, \langle \text{PAD} \rangle, \langle \text{SEP} \rangle, /\}$ where $\langle \text{BOS} \rangle$ is the beginning of sentence token, $\langle \text{EOS} \rangle$ the end of sentence token, $\langle \text{PAD} \rangle$ the padding token, $\langle \text{EDGE} \rangle$ is the token indicating an edge between two vertices and, $\langle \text{SEP} \rangle$ and "/" are separator tokens. Each sequences describes the graph by a list of all the edges followed by two randomly sampled vertices and the shortest path between these latter (see Figure 2). All the graphs are directed and sampled according to the Erdös-Rényi model, with $n$ vertices and probability $p$ for each edge to exist. We vary $n \in \{25, 30, 50, 40, 45, 50, 55\}$ and set $p$ such that the average length of the shortest path is 3.5. Each train/test pair corresponds to *one* value of $(n, p)$, we do not mix graph sizes.

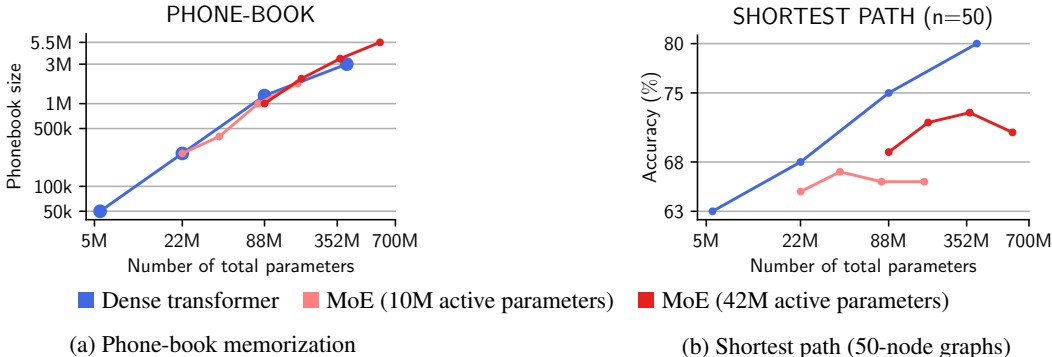

Figure 3: Illustration of the phone-book task for closed-book retrieval. The model is first trained to memorize a phone-book (illustrated on the right). Then, we randomly select a name in the phone-book (in green) and ask the model to return their phone number (in purple) without access to the phone-book.

(a) Phone-book memorization

(b) Shortest path (50-node graphs)

Figure 4: **(a) Phone-book memorization**: We train a series of dense transformers and MoEs on phone-books of varying sizes and then evaluate their memorization capacity. We report the maximal phone-book size where the model obtains more than 90% accuracy. The maximal phone-book size correlates with the total (and not active) number of parameters. **(b) Shortest path (total parameters)**: We train models to find the shortest path in 50-node graphs and report the test accuracy. Here, increasing the number of experts provides limited improvements and the performance rather correlates with the number of active parameters.

**Phone-book task.** Our token space $\mathcal{V}$ is of size 39 and made of the alphabet letters, digits and special tokens: $\mathcal{V} = \{a, \ldots, z, 0, \ldots, 9, \langle \text{BOS} \rangle, \langle \text{EOS} \rangle, \langle \text{SEP} \rangle\}$. We generate phone-books where the names consist of 5 letters and the phone numbers of 8 digits (see Figure 3). We ensure that both the names and numbers are unique.

**Datasets.** For the graph experiments, the training set size is 1e6 and the test set consists of 1e3 held-out examples that are sampled from the *same* distribution as the training examples. For the phone-book experiments, we vary the training set size over $\{1e5, 5e5, 1e6, 1.5e6, 2e6, 2.5e6, 3e6\}$ and the test set consists of 1e3 queries from the training set.

**Optimization.** We use the AdamW optimizer (Loshchilov et al., 2017) with a weight decay equal to 0.1. We sweep the learning rate over $\{5e-5, 1e-4, 5e-4, 1e-3\}$, the number of epochs over $\{2, 5, 10, 15\}$, and set the maximal possible batch size among $\{8, 16, 32\}$. We use a warmup during the 20% first training steps and a linear decay scheduler. All models are trained by next-token prediction. In the graph task, we apply a mask on the input instance so that we only penalize the model whenever it makes a mistake on the labels (and not on the inputs and labels jointly). In the phone-book experiment, we do not apply any masking.

**Evaluation.** For each task we compute the exact accuracy, i.e. we count the generation as correct only if it fully matches the ground truth. For the phone-book task, we report the size of the maximal phone-book where we observe at least 90% exact accuracy.

## 4.2 MEMORIZATION: TOTAL PARAMETERS PREDICT PERFORMANCE

We train dense transformers and MoEs on phone-books of different sizes and at test time, evaluate whether they memorize the phone number of some names. Figure 4a reports the maximal phone-book size where the model manages to get an accuracy greater than 90%. This gives us an estimate of the memorization capacity of the model.

The findings are clear: no matter the number of active parameters, MoEs match the performance of dense transformers with the same number of total parameters. This suggests that MoEs are able to effectively leverage the extra parameters in additional experts by routing tokens to the experts that contain the necessary information from the training corpus. This scaling is remarkable in this case since it even holds when we are only routing to 2 out of 64 experts. For instance, we find that an MoE model with only 42M active parameters outperforms a dense model with 10x as many parameters. This type of impressively efficient memorization capacity may be a major reason behind the success of MoE architectures.

### 4.3 REASONING: TOTAL PARAMETERS DO NOT PREDICT PERFORMANCE

We train dense transformers and MoEs on the shortest path task and then query the models to find the shortest paths in novel, held-out graphs. Figure 4b reports the performance on graphs with 50 nodes with respect to their number of total parameters. Contrary to the phone-book experiment, increasing the number of experts does not consistently improve the performance of MoEs. Essentially, we find that active parameters rather than total parameters is a better predictor of performance for these reasoning tasks.

To connect back to the theory from Section 3, note that active parameters is directly determined by the width of the network since we always route to exactly 2 experts and fix the depth. Thus, these results corroborate the theory by showing that width (i.e. active parameters) determines the performance on these graph reasoning problems and that increasing the number of experts is not helpful. In Section 5, we will further corroborate this idea through evaluation of pre-trained models on commonsense and math reasoning benchmarks.

## 5 PRE-TRAINED MODELS

In this section, we pre-train dense transformers and MoEs and compare their performance on standard math and natural language benchmarks. We break the downstream tasks into those that require more memorization and those that require more reasoning. Here, memorization refers to recall in that we measure the ability of the models to retrieve real-world facts, The memorization-intensive tasks test for "world knowledge" and consist of benchmarks like TriviaQA (Joshi et al., 2017). We break the reasoning-intensive tasks into two subcategories: one for natural language reasoning tasks like WinoGrande (Sakaguchi et al., 2021) and another for mathematical reasoning tasks like Hendrycks-MATH (Hendrycks et al., 2021). These tasks may be seen as real-world analogs of the stylized phone-book and shortest path tasks studied in Section 4.

We observe that performance on world-knowledge tasks is governed by the total number of parameters while performance on reasoning tasks depends more on the number of active parameters (Figure 1). Additionally, we conduct an experiment that indicates memorization from MoEs may be harming reasoning performance since there is a larger gap between train and test accuracy for MoEs than dense models at fixed total parameters (Figure 5). Finally, we conduct an ablation where we compare models at fixed validation perplexity rather than model size. We find that MoEs perform better on world knowledge tasks and similarly on reasoning tasks compared to dense models (Figure 6).

### 5.1 SETUP

**Architecture.** We train dense transformers and MoEs using the OLMoE codebase (Muennighoff et al., 2024). We set the number of layers $L = 20$ and vary the width $d \in \{256, 512, 1024, 2048, 4096\}$ for dense transformers and $d \in \{256, 512, 1024\}$ for MoEs. Similarly to Muennighoff et al. (2024), we consistently set the intermediate dimension in the FFN/MoE blocks to $d$ (and not $4d$). For MoEs, we vary the number of experts $E \in \{8, 16, 32, 64\}$. For the specific case of width 256, we also train a MoE with 256 experts because its parameter count approximately matches the one of a width-2048 dense model and thus, we can compare the downstream performance of the two models. We use top-2 token-choice routing, without token dropping which is implemented in the dMoE function from the Megablocks package (Gale et al., 2023). We leave the study of MoEs trained with other routing mechanisms for future work.

**Training hyperparameters.** We use the AdamW optimizer (Loshchilov et al., 2017) with a weight decay equal to 0.1. We set the learning rate to 0.001, train on 63B tokens (60k steps) with batch size 512 and sequence length of 2048. We use warmup during the 20% first training steps and a linear decay scheduler. We train our models using FSDP (Zhao et al., 2023).

**Pre-training datasets.** We train two collections of models, one collection on natural language and another one on math. The natural language dataset is a mixture constituted of FineWeb-edu (Penedo et al., 2024), Cosmopedia (Ben Allal et al., 2024), Wikipedia and the training sets of the downstream tasks we evaluate on. The math dataset is a mixture made of Proof-Pile 2 (Azerbayev et al., 2023) and instruction datasets such as OpenMathInstruct (Toshniwal et al., 2024) and MetaMathQA (Yu et al., 2023). Each of the two training mixture approximately totals 65B tokens. A precise description of the training mixtures can be found in Appendix B.

**Evaluation.** We measure the validation perplexity on 5,000 held-out sequences sampled from the training distribution. And we evaluate our models on a series of natural language and math benchmarks. Explicitly, we divide them into three categories:

– World-knowledge tasks: TriviaQA (Joshi et al., 2017), Natural Questions (Kwiatkowski et al., 2019), HotpotQA (Yang et al., 2018), WebQuestions (Berant et al., 2013), ComplexWebQuestions (Talmor & Berant, 2018).
– Commonsense tasks: ARC-C and ARC-E (Clark et al., 2018), CommonsenseQA (Talmor et al., 2018), HellaSwag (Zellers et al., 2019), OpenbookQA (Mihaylov et al., 2018), PIQA (Bisk et al., 2020), SciQ (Welbl et al., 2017), SIQA (Sap et al., 2019), WinoGrande (Sakaguchi et al., 2021).
– Math: SVAMP (Patel et al., 2021), GSM8k (Cobbe et al., 2021), GSM-Hard (Gao et al., 2023), Hendrycks-MATH (Hendrycks et al., 2021), Minerva-MATH (Lewkowycz et al., 2022).

In all our experiments, we plot the average accuracy for each of these three categories. We report the corresponding per-task performance in Appendix C.

## 5.2 RESULTS

**Experts improve memorization more than reasoning.** We observe that the conclusions from our theoretical results and synthetic experiments also hold when pre-training and evaluating language models on natural language and math. In Figure 1a, we report the accuracy of our models with respect to the number of *total* parameters. All the lines in the plot approximately coincide which implies that regardless of the number of active parameters, MoEs can effectively use their routing to leverage all of their parameters to solve memory-intensive tasks. On the other hand, on commonsense and math benchmarks (Figures 1b,1c) we find that MoEs do not reach the performance of dense models with the same number of total parameters. This indicates that for these reasoning tasks, increasing the dense model width is more effective that adding experts.

**On math tasks, MoEs display a higher train-test gap than dense models, suggestive of memorization.** We provide additional evidence that memorization occurs in pre-trained MoEs by considering the generalization gap. In

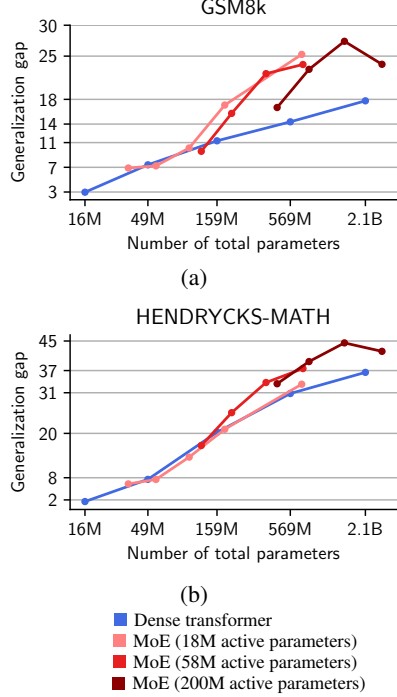

Figure 5: Generalization gap when the test set is GSM8k (a) and Hendrycks-MATH (b).

Figure 5 we select 6,319 random problems from the OpenMathInstruct dataset, which is part of the training mixture data. More precisely, we pick 5,000 Hendrycks-MATH like examples and 1,319 GSM8k-like examples to ensure that the number of training examples matches with the corresponding number of examples in GSM8k and Hendrycks-MATH test sets. We then report the *generalization gap*, which is the gap between the accuracy on training examples and test examples. While both dense transformers and MoEs make a *single* pass on the OpenMathInstruct dataset, Figure 5 shows that at scales beyond 159M parameters, MoEs suffer from a more significant generalization gap than

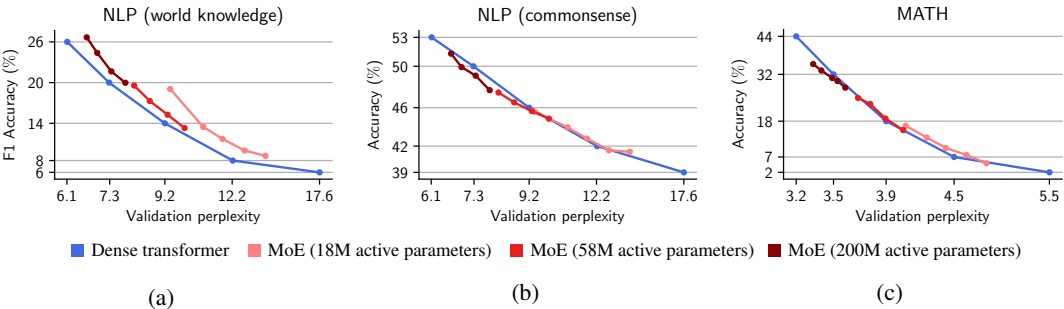

Figure 6: (a) On world knowledge benchmarks, MoEs consistently outperform dense transformers in downstream performance when fixing the validation perplexity. (b-c) In reasoning benchmarks, dense transformers perform about the same as MoEs at a fixed validation perplexity. MoEs can achieve these perplexities with less active parameters, but may require substantially more total parameters.

dense transformers. This is suggestive that MoEs are more prone to overfit to the problems they have been pre-trained on than dense models.

**MoE models excel at world knowledge tasks but match dense models in reasoning when perplexity is fixed.** Finally, we focus on the relationship between validation perplexity and downstream performance in Figure 6. Rather than comparing models by their parameter count, we can compare them based on how well they fit the training distribution as measured by validation perplexity. Even though two models may have the same perplexity, they will have learned different functions. The question is then if we can see any high level patterns in which types of functions a particular model class is more likely to learn. Figure 6a shows that at a fixed perplexity, the MoE models outperform the dense models on world knowledge tasks.

On the other hand, Figures 6b and 6c show that MoEs and dense models perform about the same on the reasoning tasks at fixed validation perplexity. Note that both dense Transformers and MoEs are trained with the objective of minimizing the perplexity (or loss). However, there could be multiple strategies to achieve the same loss, e.g. by memorizing pieces of data or by improving "reasoning" capabilities. Which strategy the model prefers is determined by the implicit bias of the architecture. In the experiments, we see that for memorization / factual recall, MoEs achieve better accuracy for the same pereplexity value, suggesting that they "prioritize" memorization over other capabilties.

## 6  DISCUSSION

In recent years, scaling up the number of parameters in Transformers has been the dominant approach for improving performance on language modeling. A standard Transformer of dimension $d$ and sequence length $L$ has number of parameters which scales with $O(d^2)$, and run-time that scales with $O(d^2L^2)$. Improving the efficiency can either attempt to reduce the dependence on $L$ or $d$. Sub-quadratic attention variants attempt to improve dependence on $L$ (Katharopoulos et al., 2020; Peng et al., 2023; Fu et al., 2022; Gu & Dao, 2023), while MoEs attempt to improve dependence on $d$ by scaling the number of parameters without scaling the dimension of the model.

This paper illuminates the costs and benefits of this reduced dependence on $d$. We show that for some reasoning-intensive tasks increasing the dimension $d$ is inevitable, and scaling the computation with $O(d^2)$ seems unavoidable. This remains true regardless of the different design choices in the MoE architecture and is backed up empirically. There is increasing interest in developing non-MoE models with sub-quadratic dependence on $d$, using some structural assumptions on the weight layers (Kamalakara et al., 2022; Dao et al., 2021; 2022; Fu et al., 2024), which could provide an alternative.

On the other hand, we find that MoEs are highly effective at knowledge intensive tasks. They are able to much more efficiently memorize facts than dense models with a similar number of active parameters, even matching the performance of dense models with the same number of total parameters. This suggests that MoEs are valuable as memorization machines and perhaps this particular capability can be leveraged while relying on other architectures for more reasoning-intensive tasks.

## ACKNOWLEDGEMENTS

We thank Cyril Zhang for helpful discussions and Max Shad for his support while running the experiments. Kempner Institute computing resources enabled this work. Samy Jelassi acknowledges funding supported by the Center of Mathematical Sciences and Applications. Alex Gu is supported by the National Science Foundation (NSF) Graduate Research Fellowship under Grant No. 2141064. David Alvarez-Melis acknowledges support from the Kempner Institute, the Aramont Fellowship Fund, and the FAS Dean's Competitive Fund for Promising Scholarship. This work has been made possible in part by a gift from the Chan Zuckerberg Initiative Foundation to establish the Kempner Institute for the Study of Natural and Artificial Intelligence; support from the Office of Naval Research under award N00014-22-1-2377, and the National Science Foundation Grant under award #IIS 2229881.

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

## A    LIMITATIONS AND FUTURE WORK

While we provide substantial experiments on a wide range of tasks, many of these tasks are not pure memorization nor pure reasoning. We study these two extreme cases to better convey our message but it would be interesting in future work to understand the difference between dense and sparse models on tasks that mix both memorization and recall. Our pre-trained models have up to $\leq 2.1$B parameters, but we recognize that large scale MoEs like Mixtral (Jiang et al., 2024), DeepSeek-V2 (Dai et al., 2024), and others have orders of magnitude more parameters. We hypothesize that our results would still be meaningful at larger scales due to the strong theoretical underpinning, but it is not guaranteed. Moreover, as suggested above, it would be an interesting direction for future work to propose new architectures with reduced $d$ dependence that can get the best of both worlds and solve reasoning and memorization tasks.

## B    DETAILS ON THE PRE-TRAINING DATASETS

In Section 5, we pretrain two collections of models, one on "natural language" and the other on "math". Here, we give a precise breakdown of our training mixtures. We start with the "natural language" training mixture that totals 64B tokens:

- 37B tokens from Fineweb-edu dedup (Penedo et al., 2024).
- 14B tokens from Cosmopedia (Ben Allal et al., 2024).
- 12B tokens from Wikipedia (we loop over Wikipedia 3 times).
- 1B tokens from the training set of the downstream tasks we test on. We create 3 copies of each of these to increase their presence in the mixture. The presence of these datasets is pretty important as argued in Allen-Zhu & Li (2023) so that the model is familiar with the downstream tasks at test time.
    * ComplexWebQuestions training set (Talmor & Berant, 2018)
    * HotPotQA training set (Yang et al., 2018)
    * Natural Questions training set (Kwiatkowski et al., 2019)
    * TriviaQA training set (Joshi et al., 2017)
    * WebQuestions training set (Berant et al., 2013)
    * ARC-Easy and ARC-Challenge training sets (Clark et al., 2018)
    * Hellaswag training set (Zellers et al., 2019)
    * OpenBookQA training set (Mihaylov et al., 2018)
    * PIQA training set (Bisk et al., 2020)
    * SciQ training set (Welbl et al., 2017)
    * SIQA training set (Sap et al., 2019)
    * Winogrande training set (Sakaguchi et al., 2021)

Our "math" training mixture that totals 66B tokens gathers:

- 55B tokens from Proof-Pile 2 (Azerbayev et al., 2023) that contain AlgebraicStack (11B), OpenWebMath (Paster et al., 2023) and ArXiv (29B).
- 2B tokens from OpenMathInstruct-1: we select the instances with a correct answer from the training set (Toshniwal et al., 2024)
- 7B tokens from DeepMind math (Saxton et al., 2019)
- 2B tokens from the following instruction-like datasets:
    * Math-Orca (Mitra et al., 2024)
    * TinyGSM (Liu et al., 2023) (we only select 1 million examples from there).
    * StackMathQA (Zhang, 2024)
    * MAmmoTH2 (Yue et al., 2024) (we only select the mathstackexchange subset).
    * NuminaMath-CoT (NuminaMath, 2024) (duplicated 3 times)
    * MetaMathQA (Yu et al., 2023) (duplicated 3 times)

## C ADDITIONAL EXPERIMENTS

In all our experiments in Section 5, we report the average accuracy performance obtained by our pre-trained models on respectively world knowledge, commonsense and math benchmarks. Here, we provide the results per task. In Subsection C.1, we display for each task, the downstream performance on a per parameter basis (similar to Figure 1) and in Subsection C.2, we plot for each task, the downstream performance on a per validation perplexity basis (similar to Figure 6).

### C.1 DOWNSTREAM PERFORMANCE ON A PER PARAMETER BASIS

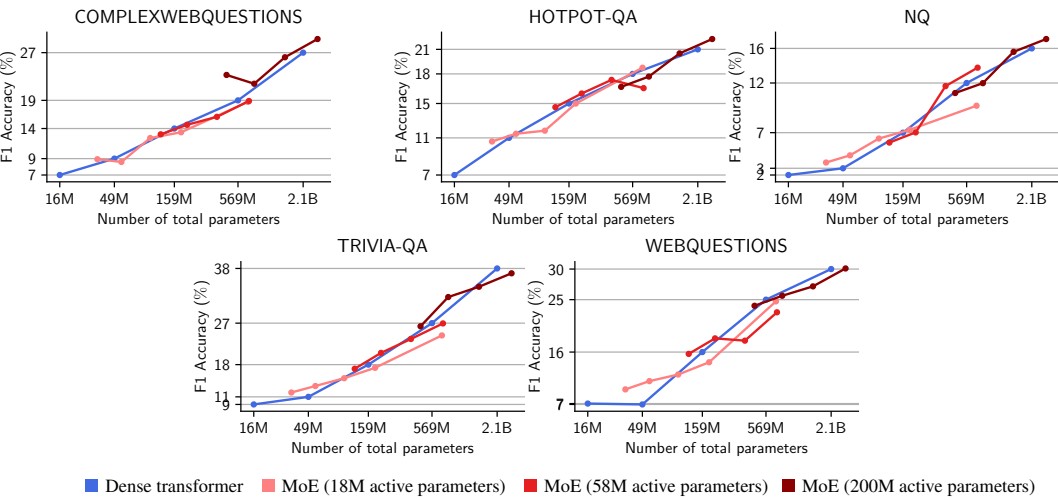

Figure 7: Downstream performance on the world knowledge tasks with respect to the total number of parameters of the models.

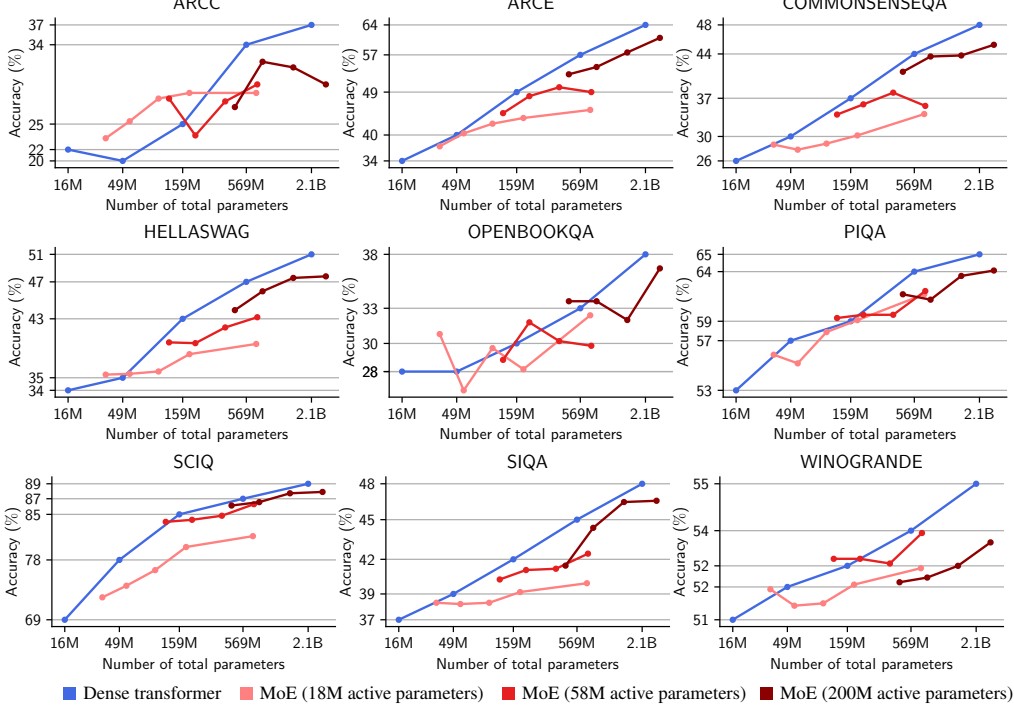

Figure 8: Downstream performance on the commonsense tasks with respect to the total number of parameters of the models.

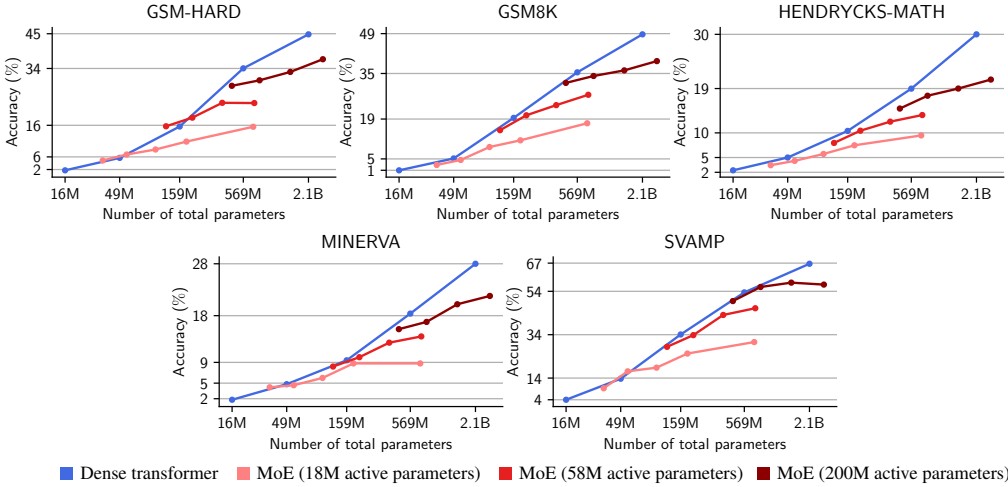

Figure 9: Downstream performance on the math benchmarks with respect to the total number of parameters of the models.

## C.2 DOWNSTREAM PERFORMANCE ON A PER VAL PERPLEXITY BASIS

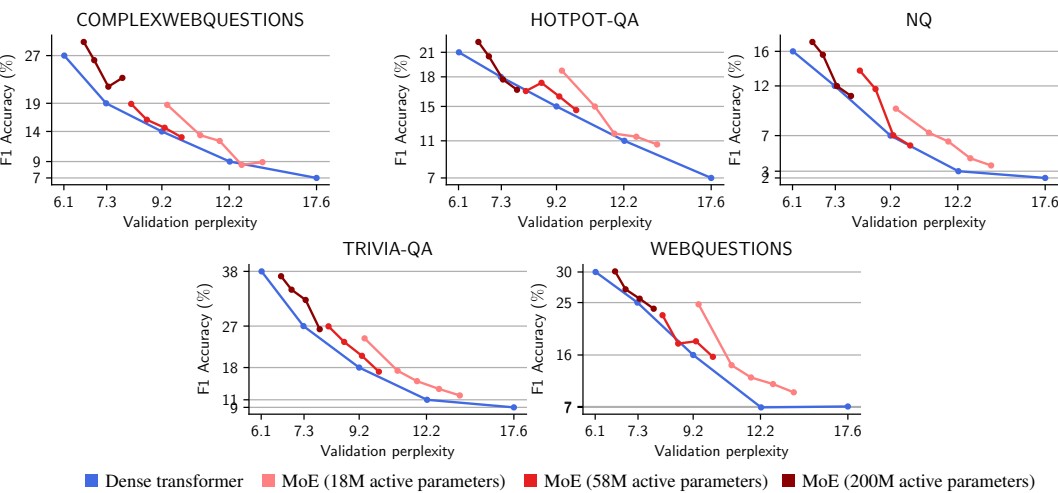

Figure 10: Downstream performance on the world knowledge tasks with respect to the validation perplexity.

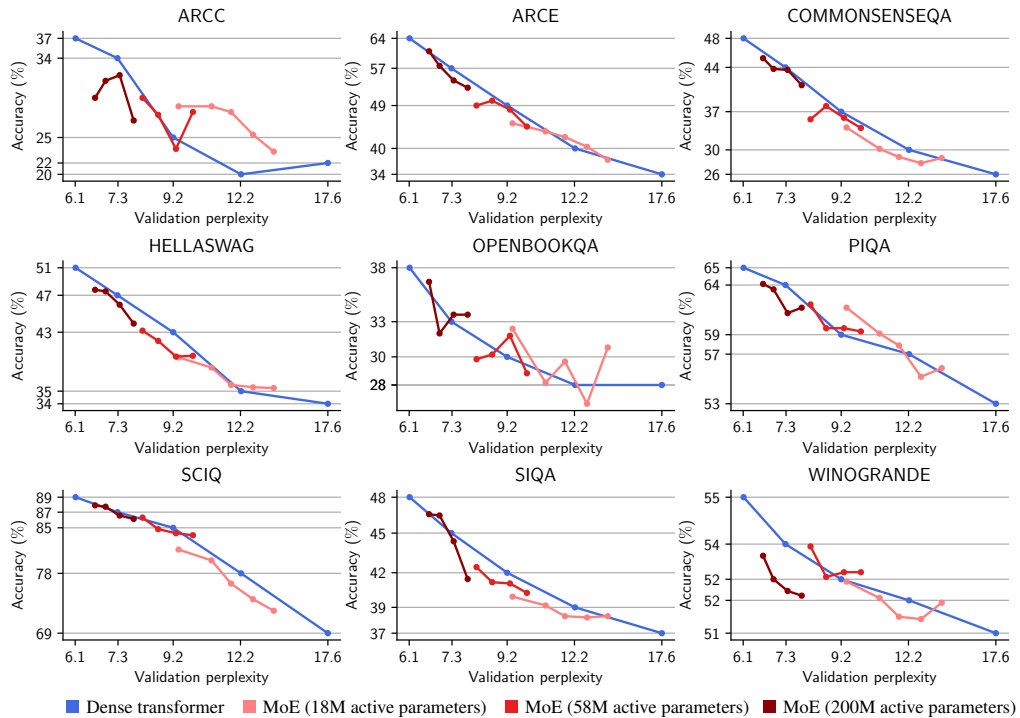

Figure 11: Performance on the commonsense tasks with respect to the validation perplexity.

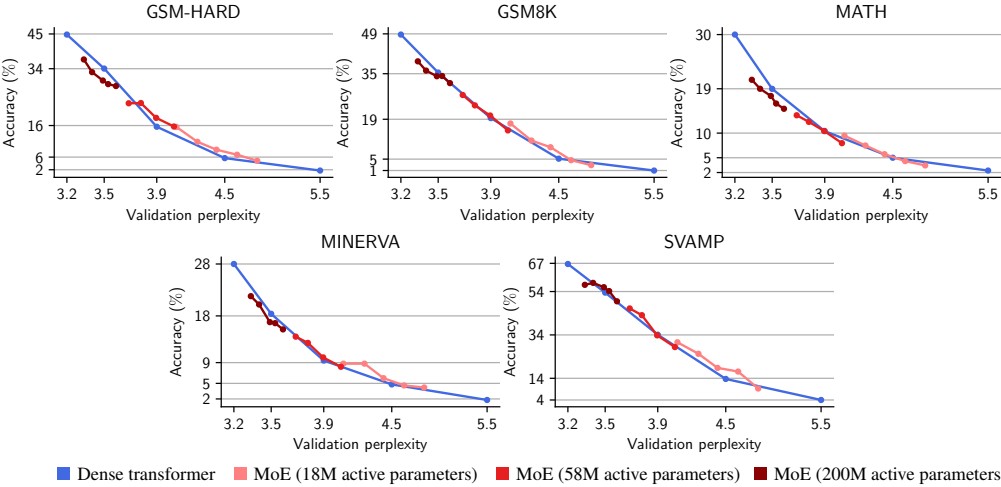

Figure 12: Downstream performance on the math benchmarks with respect to the validation perplexity.

# D PROOFS

## D.1 REASONING PROOFS

**Definition D.1** (Set-disjointness task). Set disjointness is the following task: given two inputs $A, B \in \{0, 1\}^r$ for some $r \in \mathbb{N}$, compute $\max_i A_i B_i$.

Set-disjointness can be thought of as follows: Alice and Bob are given sets $A$ and $B$ respectively. Their objective is to determine whether they have any overlapping items in their sets.

**Lemma D.2** (Equivalence of set-disjointness and length-2 path). *The set-disjointness task is equivalent to the length-2 path task.*

*Proof.* ( $\implies$ ): Given an instance of set-disjointness, we can encode it into a length-2 path problem. Denote every item $i$ as a vertex. Denote two extra vertices as $A, B$, corresponding to Alice and Bob. For every element $i$ that Alice has, draw an edge between $A$ and $i$. For every element $i$ that Bob has, draw an edge between $B$ to $i$. If and only if there are any overlapping elements, then there is a length-2 path from $A$ to $B$. The number of elements because the number of vertices that do not belong to Alice or Bob.

( $\impliedby$ ): Consider an instance $G = (V, E), s, d$ of length-2 path, where $s$ is the source vertex and $d$ is the sink vertex. For all vertices with an edge with $s$, put this element into Alice's set of elements. For all vertices with an edge with $d$, put this element into Bobs's set of elements. If and only if there is a length-2 path, then Alice and Bob's sets are overlapping. Then, $r$ is the number of vertices. $\square$

**Lemma D.3** (Communication complexity lower-bound on concatenated outputs). *For some sequence length, fix two disjoint subsets $A, B \subset [N-1]$, and consider a single-layer transformer $f \in \text{Transformer}_{m,H,1}^N$ with $O(\log N)$-bit precision that solves set disjointness for any input $X$ where $X_A$ is a function of Alice's input $a \in \{0,1\}^r$, $X_B$ is a function of Bob's input $b \in \{0,1\}^r$, and $X_{[N]\setminus(A\cup B)}$ is fixed regardless of $a, b$. Then, $f$ has width satisfying $mH = \Omega(r/\log N)$.*

*Proof.* By re-writing the following, the remainder of the proof from Sanford et al. (2024) still holds.

$$\text{DISJ}(a, b) = \psi\Big( \big[\text{softmax}\big(\phi(x_N)^\top Q_h K_h^\top \phi(X)\big)\phi(X)v_h\big]_{h\in[H]} \Big).$$

This is because we may still use the same definition for $Z_{h,S}, L_{h,S}$ as in the proof. Hence, this concludes the proof. $\square$

### D.1.1 PROOF OF THEOREM 3.2

We restate the corollary.

**Theorem D.4** (Theorem 3.2). *For some input sequence $G = (V, E)$, fix two disjoint subsets $A, B \subset [N-1]$, and consider a single-layer transformer $f \in \text{Transformer}_{m,H,1,K}^N$ with $O(\log N)$-bit precision that solves length-2 path for any input $X$ where $X_A$ is a function of edges with the source $s$, $X_B$ is a function of edges with the destination $d$. Then, $f$ has width satisfying $mH = \Omega(|V|/\log N)$.*

*Proof.* The proof outline is as follows:

1. Adapt Lemma 39 (Sanford et al., 2024) to support concatenation instead of addition from different attention heads.

2. The lower bound with concatenation holds for length-2 path because set-disjointness and length-2 path are equivalent.

3. Extend the result to sparse transformers.

We complete the first step with Lemma D.3. We complete the second set due to Lemma D.2. It remains to show that a router function also yields the same lower bound. We show that Lemma 39 of Sanford et al. (2024) can be generalized to the case in which $\psi$ is applied according to a routing function. Specifically, consider a top-1 routing function $r : \mathbb{R}^m \to [K]$, and $K$ element-wise functions $\psi_1, \ldots, \psi_K : \mathbb{R}^m \to \mathbb{R}$. For shorthand, define:

$$Y(X_N) = \left[\text{softmax}\left(\phi(x_N)^\top Q_h K_h^\top \phi(X)\right) \phi(X) v_h\right]_{h \in [H]},$$

which is the output of the attention head prior to applying the element-wise transformation. Next, we define $f(X_N)$ as the output when the router function $r$ is used to select $\psi_i$.

$$f(X_N) = \sum_{i \in K} \mathbf{I}\{r(Y(X_N)) = i\}\psi_i(Y(X_N)).$$

Because the lower bound does not place any restrictions on the function $\psi$ and rather argues a communication-complexity lower bound due to information from $Y(X_N)$, the lower bound also holds for a routing function. $\square$

### D.1.2 PROOF OF THEOREM 3.3

We re-state Theorem 3.3 and give its proof.

**Theorem D.5** (Theorem 3.3). *For sequence length $N$, there exists $f \in \text{Transformer}_{m,1,1}^N$ with $O(\log N)$-bit precision and width $|V|$ that solves length-2 path for any input $X$.*

*Proof.* Tokens are elements in $\mathcal{V} = V \cup \{0\} \times V \cup \{0\}$. The input is as follows: for vertex $i$, if the source shares an edge with that vertex, then the $i$'th input value is $(s, i)$. Otherwise, it is $(s, 0)$. The first $|V|$ tokens we see correspond to edges possibly shared with the source vertex. Then, the last $|V|$ input tokens correspond to edges possibly shared with the destination vertex and share the same format as the first $r$ tokens. In between, we can have arbitrary edges $(u, v)$. We define an embedding function where $\mathbf{e}_i$ is the $i$'th standard basis vector in dimension $r$.

$$\phi : \mathcal{V} \to \mathbb{R}^{|V|}$$

$$(u, v) \mapsto \begin{cases} \mathbf{e}_i & \text{if } i > 0 \text{ and } u = s \text{ or } u = v \\ \mathbf{0} & \text{if } i = 0. \end{cases}$$

Next, we define $V_h \in \mathbb{R}^{|V| \times |V|}$ to be the identity matrix, and $Q_h, V_h \in \mathbb{R}^{|V| \times |V|}$ both to have 0 everywhere. Consequently, the attention matrix is given by:

$$\left(\begin{bmatrix} 1/|V| & \cdots & 1/|V| \\ \vdots & \ddots & \\ 1/|V| & & 1/|V| \end{bmatrix} \phi(X)\right)_{j,i} = \begin{cases} 2/|V| & \text{if there is a path through } i \\ 1/|V| & \text{if one target vertex shares an edge with } i \\ 0 & \text{otherwise.} \end{cases}$$

For any entry that exceeds $\frac{1}{|V|}$, the correct answer is there is a length-2 path. Hence, any thresholding function which achieves this separation suffices. Provided that any rounding $\frac{1}{|V|}$ to $\tilde{c}$ so that it is represented with $O(\log N)$ is sufficient to separate between $\tilde{c}$ and $2\tilde{c}$, then $O(\log N)$ bits are sufficient. $\square$

### D.2 MEMORIZATION PROOFS

Assume that $K$ is a power of 2, and let $m = m_0 + \log(K)$. We associate each expert $i$ with a vector $\mathbf{v}_i \in \{\pm 1\}^{\log(K)}$ and choose $\mathbf{r}_i = (v_1, \ldots, v_{\log(K)}, 0, \ldots, 0) \in \mathbb{R}^m$.

**Lemma D.6.** *For some $c > 0$, and some input $x \sim \mathcal{N}(0, cI_m)$, the probability of routing to expert $i$ is $1/K$ for all $i$.*

*Proof.* By construction, we choose the expert as follows:

$$i = \operatorname*{argmax}_{j \in [K]} v_j^\top x.$$

The solution above must be the one whose signs match those of $x$. Because the entries of $x$ are 0-mean, this implies that the probability of routing to any particular expert is $1/K$. $\square$

**Lemma D.7.** *Fix* $\delta \in (0,1)$ *and some expert* $j$. *With probability at least* $1 - \delta$, *the number of examples routed to* $j$ *is at most* $2N/K$, *given* $N \geq \frac{K^2 \ln(1/\delta)}{2}$ *examples from* $\mathcal{N}(0, cI_m)$.

*Proof.* The proof relies on a Hoeffding bound: for bounded independent random variables $X_1, \ldots, X_N$ such that $a \leq X_i \leq b \ \forall i$, and $S_N = \sum_{i=1}^{N} X_i$, we have:

$$\mathbb{P}[S_N - \mathbb{E}[S_N] \geq \mathbb{E}[S_N]] \leq \exp\left\{-\frac{2\,\mathbb{E}[S_N]^2}{N(b-a)^2}\right\}.$$

In our case, let $X_i = \mathbf{1}\{$sample $i$ is routed to expert $j\}$, for some fixed $j \in [K]$. Note that $\mathbb{P}[X_i = 1] = 1/K$ due to Lemma D.6 and as such $E[S_N] = N/K$. Further, $X_i \in [0, 1]$. Hence we obtain:

$$\mathbb{P}\left[S_N \geq \frac{2N}{K}\right] \leq \exp\left\{-\frac{2(N/K)^2}{N}\right\}$$

$$\leq \exp\left\{-\frac{2N}{K^2}\right\}.$$

In order to upper-bound this probability with $\delta$, we obtain:

$$\delta \geq \exp\left\{-\frac{2N}{K^2}\right\}$$

$$\implies \ln\left(\frac{1}{\delta}\right) \leq \frac{2N}{K^2}$$

$$\implies \frac{K^2 \ln(1/\delta)}{2} \leq N.$$

$\square$

**Lemma D.8.** *Fix* $\delta \in (0,1)$, $K \in \mathbb{N}$ *and the routing function as described in Lemma D.6. Given* $n \geq \frac{K^2 \ln(1/\delta)}{2}$ *samples with embedding dimension* $m$. *For every expert* $i$, *with probability at least* $1 - \delta$, *there exists an MLP of width* $\tilde{O}(n/mK)$ *that correctly classifies the examples routed to this expert.*

*Proof.* We begin with a result from Daniely (2020). Consider $h$, a depth-2 network with $q$ neurons, an activation function $\phi : \mathbb{R} \to \mathbb{R}$, which is $O(1)$-Lipschitz, piecewise twice-differentiable, and satisfies $\mathbb{E}_{X \sim \mathcal{N}(0,1)}[\phi'(X)] = 0$, weights $\mathbf{a}_i \in \{\pm 1\}$ such that $\sum_{i=1}^{q} a_i = O(\sqrt{q})$. Such a network, defined by $W$ and $a$, computes the function:

$$h_{W,a}(x) = \frac{1}{\sqrt{q}} \sum_{i=1}^{q} a_i \phi(\langle w_i, x \rangle).$$

For our proof we consider $\phi$ to be the absolute value function, and later show how to obtain an MLP with ReLU activations.

Consider the set of samples $((x_1, y_1), \ldots, (x_N, y_N))$, where $x_i \sim \mathcal{N}(0, I_M)$ and $y_i \in \{\pm 1\}$ is a Rademacher random variable. The objective is to find a $W^*, a^*$ such that $y_i h_{W^*, a^*}(x_i) > 0 \ \forall i$. Daniely (2020) gives the following result: for $N \leq \frac{Mq}{\log^4(M)}$ and $q \geq \log^4(M)$, with probability $1 - o(1)$ there exists $h$ such that, for every $i \in [N]$, $y_i h(x) = \Omega(\ln(M))$.

We build on this result as follows: first, we assume that the attention matrix performs averaging of the input sequence, such that $x_i = \frac{1}{N} \sum_{j=1}^{N} (X^i)_j$ where where $X^i$ is the $i$-th input sequence to memorize. Assuming the inputs are also Gaussian, then so is the average. Hence, $x_i \in \mathbb{R}^m$ and $x_i \sim \mathcal{N}(0, cI_m)$ for $c = \frac{1}{L}$.

Next, we fix the number of experts to be $K$. We fix a router function $r : \mathbb{R}^{\log K} \to [K]$ such that, on average, each expert must memorize $n/K$ sequences, and with high probability, each expert must memorize at most $2n/K$ sequences. We use the construction from Lemma D.6 where the router only acts on the first $\log K$ entries of each vector. Hence, we have $K$ MLPs, each tasked with memorizing

at most $\frac{2n}{K}$ input-output values. We use the network construction from Daniely (2020) to only the last $m - \log K \geq m/2$ coordinates, which are not used by the router, so that their distribution is independent and Gaussian.

Hence, with $n \leq \frac{K}{2} \frac{(m-\log K)q}{\log^4(m-\log K))} \leq \frac{K}{4} \frac{mq}{\log^4(m-\log K))}$ and $q \geq \log^4(m - \log K)$, with high probability, $y_i h_j(x_i) = \Omega(\ln(m))$, where $h_j$ is the expert to which $x_i$ is routed. It follows that width

$$q \geq \frac{4n \log^4(m - \log K)}{mK}$$

is sufficient for each individual expert, and we assume that $\frac{4n}{mK} \geq 1$.

In order to obtain a MLP with a ReLU activation defined as $\sigma(x) = \max\{0, x\}$, we use the fact that $|x| = \sigma(x) + \sigma(-x)$. This is because absolute value is a valid activation function in Daniely (2020), but ReLU is not. However, doubling the width of each MLP is sufficient to obtain MLPs with ReLU activations instead, i.e.:

$$q \geq \frac{8n \log^4(m - \log K)}{mK}.$$

This is done by taking a solution which is of form:

$$h_{W,a}(x) = \frac{1}{\sqrt{q}} \sum_{i=1}^{q} a_i |\langle w_i, x \rangle|,$$

and converting it to the form:

$$h'_{W,a}(x) = \frac{1}{\sqrt{q}} \sum_{i=1}^{q} a_i ((\sigma \langle w_i, x \rangle) + (\sigma \langle -w_i, x \rangle)).$$

The router has $K$ vectors each of dimension $m$. Consequently, we need $O(Km)$ parameters to store the router. Each expert has width $\frac{8n \log^4(m-\log K)}{mK} = \tilde{O}\left(\frac{n}{mK}\right)$ and $\tilde{O}\left(\frac{n}{K}\right)$ parameters.  □

**Corollary D.9.** *Let $\delta \in (0, 1)$, and fix $K > 1$. For $n \geq \frac{K^2 \ln(K/\delta)}{2}$, with probability at least $1 - \delta$, there exists a sparse transformer $s$ with $K$ experts such that $y_i s(x_i) = \Omega(\ln m)$. It has $\tilde{O}(n + Km)$ parameters and $\tilde{O}(n/K + Km)$ active parameters.*

*Proof.* For each expert, we apply the result from Lemma C.8 with $\delta' = \delta/K$. Hence, for every expert, with probability at most $\delta/K$, there does not exist an MLP of width $\tilde{O}(n/mK)$ which memorizes its samples. By a union bound, this implies that with probability at most $\delta$, at least one of the experts is not able to memorize its samples. Hence, with probability at least $1 - \delta$, all of the experts are able to perform memorization on their given samples. In total, this implies we will use $\tilde{O}(n + K)$ parameters to store the entire mixture of expert network. The number of active parameters is $\tilde{O}(n/K + K)$.  □

**Lemma D.10** (Bound on $\ell_2$-norm of vector from $\mathcal{N}(0, cI_m)$). *Let $x \sim \mathcal{N}(0, cI_m)$ for some $c > 0$, and let $\delta \in (0, 1)$. Then, with probability at least $1 - \delta$, there exists a constant $c_{\delta,m} > 0$, such that*

$$\|x\|_2 \leq c_{\delta,m} = \sqrt{c \left( m + 2\sqrt{m \ln\left(\frac{1}{\delta}\right)} + 2\ln\left(\frac{1}{\delta}\right) \right)}.$$

*Proof.* Each component $x_i$ of $x$ is distributed as $x_i \sim \mathcal{N}(0, c)$. We can express $x_i$ as $x_i = \sqrt{c}\, z_i$, where $z_i \sim \mathcal{N}(0, 1)$. Then,

$$\|x\|_2^2 = \sum_{i=1}^{m} x_i^2 = c \sum_{i=1}^{m} z_i^2.$$

The sum $\sum_{i=1}^{m} z_i^2$ follows a chi-squared distribution with $m$ degrees of freedom. By the Laurent-Massart theorem Laurent & Massart (2000), for all $t > 0$, we have

$$\mathbb{P}\left( \sum_{i=1}^{m} z_i^2 \geq m + 2\sqrt{mt} + 2t \right) \leq e^{-t}.$$

Multiplying both sides inside the probability by $c$, we obtain

$$\mathbb{P}\left(\|x\|_2^2 \geq c\left(m + 2\sqrt{mt} + 2t\right)\right) \leq e^{-t}.$$

Setting $t = \ln\left(\frac{1}{\delta}\right)$, it follows that

$$\mathbb{P}\left(\|x\|_2^2 \geq c\left(m + 2\sqrt{m\ln\left(\frac{1}{\delta}\right)} + 2\ln\left(\frac{1}{\delta}\right)\right)\right) \leq \delta.$$

Therefore, with probability at least $1 - \delta$, we have

$$\|x\|_2 \leq \sqrt{c\left(m + 2\sqrt{m\ln\left(\frac{1}{\delta}\right)} + 2\ln\left(\frac{1}{\delta}\right)\right)}.$$

$\square$

**Lemma D.11** (Bounded bit complexity required). *With high probability, $\tilde{O}(1)$ bits are required to store each weight in the network, and $\tilde{O}(n + K)$ bits are required to store the entire network. $\tilde{O}\left(\frac{n}{K} + K\right)$ active bits are required.*

*Proof.* We show that the learning result from Daniely (2020) gives a network with bounded bit complexity. Hence, it suffices to bound the bit complexity of the initial weights and the bit complexity of the gradient update. First, we begin with $h(x) = \sqrt{\frac{1}{q}} \sum_{i=1}^{q} a_i \sigma(\langle w_i, x \rangle)$. The objective is to show there exists $\tilde{h}(x)$ with bounded bit complexity which satisfies $y_i \tilde{h}(x_i) > 0 \; \forall i$. Suppose we begin by creating bins for the weights, and replace each $w_i$ with $\tilde{w}_i$ where $\|w_i - \tilde{w}_i\|_\infty \leq \varepsilon$. Further, assume that $\|w_i\|_\infty \leq M$ for some $M > 0$. We then obtain that $\|w_i - \tilde{w}_i\|_2 \leq \sqrt{m}\varepsilon$. In addition, we state that with probability at least $1 - \delta$, $\|x\|_2 \leq c_{\delta,m}$ due to Lemma D.10. Using this, we have that:

$$
\begin{aligned}
|\sigma(\langle w_i, x \rangle) - \sigma(\langle \tilde{w}_i, x \rangle)| &\leq |\langle w_i - \tilde{w}_i, x \rangle| && (\sigma \text{ is 1-Lipschitz}) \\
&\leq \|w_i - \tilde{w}_i\|_2 \|x\|_2 && (\text{Cauchy-Schwarz}) \\
&\leq \sqrt{m}\varepsilon c_{\delta,m} \text{ w.p.} \geq 1 - \delta && (\text{Lemma D.10}).
\end{aligned}
$$

We apply this result to obtain that with probability at least $1 - \delta$,

$$
\begin{aligned}
|h(x) - \tilde{h}(x)| &\leq \frac{1}{\sqrt{q}} \sum_{i=1}^{q} |a_i| |\sigma(\langle w_i, x \rangle) - \sigma(\langle \tilde{w}_i, x \rangle)| \\
&\leq \sqrt{q}\sqrt{m}\varepsilon c_{\delta,m}.
\end{aligned}
$$

Given that $h(x) \geq O(1)$, then for some arbitrarily small constant (we use $\frac{1}{4}$) we require: $\frac{1}{4} \leq \sqrt{q}\sqrt{m}\varepsilon c_{\delta,m}$, or equivalently, $\varepsilon \leq \frac{1}{4c_{\delta,m}\sqrt{qm}}$. Because $h(x) \geq O(1)$, we use this separation of $\frac{1}{4}$ to show that $\tilde{h}(x)$ remains the same sign as $h(x)$ (however this constant can be replaced with an arbitrarily small constant so as to satisfy the requirement).

Consider $w_i^{(0)}$ to be the initialization of $w_i$ prior to the gradient step. Then, because, $M = \|w_i\|_\infty$, we have that $M = \max_i\left\{w_i^{(0)} - \eta \frac{\partial h(x)}{\partial w_i}\right\} = \max_i\left\{w_i^{(0)} - \frac{n \log m}{m} \frac{\partial h(x)}{\partial w_i}\right\}$. Assuming that $w_i^{(0)}$ is initialized with bounded $\ell_\infty$ norm of 1, then we obtain that $M \leq 1 + \frac{n \log m}{m\sqrt{q}} c_{\delta,m}$. This is due to Lemma D.10 and that

$$
\begin{aligned}
\frac{\partial h(x)}{\partial w_i} &= \frac{1}{\sqrt{q}} a_i \sigma'(\langle w_i, x_i \rangle) x_i \\
&= \begin{cases} 0 & \text{if } \langle w_i, x_i \rangle \leq 0 \\ \frac{1}{\sqrt{q}} a_i x_i & \text{otherwise}. \end{cases}
\end{aligned}
$$

Using this, with high probability, we require $\log\left(\frac{2M}{\varepsilon}\right) = \log\left(8Mc_\delta\sqrt{qm}\right)$ bits (by replacing $\varepsilon$). Hence, with high probability, we require the number of bits to be at most:

$$O\left(\log\left(\frac{n\log m}{\sqrt{m}}c_{\delta,m}^2\right)\right) = \tilde{O}(1).$$

$\square$

We restate Theorem 3.6.

**Theorem D.12** (Lower bound for dense model)**.** *Given the same task as above, a dense Transformer requires $\tilde{\Omega}(n)$ parameters to solve the memorization task.*

*Proof of Theorem 3.6.* Let $c$ be the number of bits used for encoding each parameters (and we assume that $c$ is logarithmic in the problem parameters). Denote by $\mathcal{H}$ the class of all transformers with $W$ parameters and $c$ bits per parameters. Since $\mathcal{H}$ is a finite class, where each function in the class can be encoded with $cW$ bits, we have $|\mathcal{H}| \leq 2^{cW}$. Let $X^1, \ldots, X^N \in \mathbb{R}^{n \times d}$ be the $N$ input points. Assume a $\mathcal{H}$ can solve the memorization task. Then, for every choice of $y_1, \ldots, y_N \in \{\pm 1\}$, there exists a transformer $f \in \mathcal{H}$ s.t. $f(X_i) = y_i$ for all $i \in [N]$. There are $2^N$ possible assignments for $y_1, \ldots y_N$ and therefore there are at least $2^N$ different functions in $\mathcal{H}$. So, we get $2^N \leq |\mathcal{H}| \leq 2^{cW}$ and therefore $W \geq N/c$. $\square$

**Lemma D.13** (Active parameter comparison between dense and sparse transformers)**.** *There exist cases in which the amount of active parameters required to perform memorization is less for a sparse transformer than a dense transformer.*

*Proof.* As shown, a dense transformer requires $\tilde{\Omega}(n)$ parameters (and active parameters) to perform memorization of $N$ sequences. In contrast, for $n$ sufficiently large and fixed $K > 1$, it holds that $\frac{n}{K} + K < n$, which shows that the number of active parameters required is less for a sparse transformer. $\square$

