# OpenReview forum: "Mixture of Parrots: Experts improve memorization more than reasoning"
_ICLR.cc/2025/Conference — ICLR 2025 Poster_

### Official Review · Reviewer_toS5 · 2024-10-30

**Soundness:** 4
**Presentation:** 4
**Contribution:** 4
**Rating:** 8
**Confidence:** 4

**Summary:**

This paper proposes to study the performance tradeoffs between MoE models and standard dense transformers. More specifically, the paper proposes a theoretical analysis of MoE models at reasoning that validates the fact that MoEs can better handle knowledge-intensive tasks but are inferior to traditional dense transformers on those that need generalization ability for reasoning. Both synthetic datasets and real-world dataset verify the conclusion of this paper.

**Strengths:**

1. The paper is well-written and easy to follow, the problem this paper focuses is important to the community.
2. Theoretical analysis is sound.
3. The experiments are solid, the authors use synthetic data to provide a straightforward illustration and then extend to real-world datasets. The experiments support the claims of this paper sufficiently.

**Weaknesses:**

1. The proposed claim serves as a general conclusion. Have the authors tried on other reasoning tasks? For example, logical reasoning and code generation tasks are widely used to validate the effectiveness of reasoning.
2. If the $k$ in top-k routing (k=2 in the main paper) influence the conclusion of this paper? And have the authors tried different routing machanisms?
3. The analysis focuses on depth-1, however, in the synthetic data experiments, the authors use $L=12$ transformer layers to verify. Why does the authors not choose 1-layer transformer to verify the results on synthetic data?
4. Compared to dense transformers, if there are configurations that can perform better than dense transformers theoretically?
5. I'm curious about if the MoE performance on memory tasks and reasoning tasks is robust to the initialization or distribution of parameters.

**Questions:**

I am curious to hear the authors thoughts regarding the reasoning ability of MoEs, if there exists possible future directions to mitigate the performance gap between MoEs and dense transformers? This is quite an important direction for both development of MoEs and LLM reasoning.

---

### Official Review · Reviewer_A95z · 2024-11-03

**Soundness:** 4
**Presentation:** 4
**Contribution:** 3
**Rating:** 8
**Confidence:** 4

**Summary:**

This paper investigates the difference between how performance scales with the number of experts on reasoning vs memorization tasks. Specifically, they show through theoretical results, synthetic datasets and common benchmarks that reasoning tasks scale more with the number of active parameters whereas memorization tasks scale more with the number of total parameters.

**Strengths:**

Tradeoffs between dense and sparse models is a very relevant problem as it’s a differentiator between many existing foundation models.

The authors clearly demonstrate an interesting finding that scaling experts improves knowledge/memorization tasks but has a lesser benefit on reasoning tasks (worse than dense activation-matched alternatives).

The results are very convincing, as this trend is demonstrated on theoretical tasks, synthetic tasks designed for reasoning/memorization and on popular benchmarks
- Experiments in each section seem reasonable to back up the claim!

Graphs are very clear and the overall story is very easy to follow.

**Weaknesses:**

More elaboration on what is meant by “this suggests that MoEs do have a bias towards learning functions that memorize the training data” when talking about figure 6a would be helpful:
- Wouldn’t this bias show by just having lower perplexity? Why does the perplexity-accuracy relationship show this?

Many tasks people care about require a mix of knowledge and reasoning. It would strengthen the study to do more targeted synthetic or real benchmarks designed to explicitly test the scaling properties for these tasks.

**Questions:**

Why did you choose different architectures for the synthetic and non-synthetic tasks?

Did you consider looking at tasks that are a mix of both knowledge and reasoning (ex solving crossword puzzles)? It would be interesting to see more tradeoffs in these mixed tasks between scaling depth and number of experts.

---

### Official Review · Reviewer_7TRg · 2024-11-04

**Soundness:** 2
**Presentation:** 3
**Contribution:** 2
**Rating:** 6
**Confidence:** 4

**Summary:**

This paper explores the performance of the Mixture-of-Experts (MoE) architecture in tasks requiring different capabilities, specifically memorization and reasoning. It examines how increasing the number of experts in MoEs affects performance, particularly comparing MoEs to dense transformers. The authors find that while MoEs improve memorization substantially, they fall short in reasoning tasks compared to dense transformers. The paper includes both theoretical and empirical analyses across synthetic graph problems, closed-book retrieval tasks, and benchmarks in math and natural language, ultimately concluding that MoEs do well in memory-intensive tasks but struggle with reasoning.

**Strengths:**

1. Interesting: interesting question to differentiate the performance of MoEs and dense transformers across task types, particularly highlighting the unique strengths of MoEs vs MLPs in a task oriented way
2. Clarity: The paper is clear and well-organized
3. Glad generative tasks like GSM8K and MATH were included.
4. Solid motivation of MoEs in the literature in intro.

**Weaknesses:**

Scientific Weakness
1. Insufficient Empirical Validation: The paper would benefit from a broader set of empirical evaluations beyond the synthetic tasks used. For example, adding MMLU subtasks (eg Abstract Algebra and College mathematics are good starts. Under philosophy there is a formal logic task too. Big-Bench has an Induction task and Identify Theorem) and OlympiadBench [1] could provide richer insights into MoEs’ effectiveness on real-world tasks. This would also address the issue of over-reliance on benchmarks like GSM8K and MATH, which may be too narrow. Additional experiments at a larger model scale would also help validate the generalizability of the findings.
2. Limited Real-World Applicability: Many of the toy experiments and theoretical results are not clearly connected to practical outcomes. This should be made directly and explicitly early in the paper. The assumptions made, especially regarding data distribution, need clearer justification to demonstrate their relevance in real-world applications. Without this, it’s difficult to assess if these experiments are accurate models for practical settings. A toy example that actually encompasses reasoning is needed for me to raise my score. Something that is real human reasoning and thus doesn’t have a deterministic algorithm that solves it (like random graphs or arithmetic).
3. Alternative Approaches Left Unexplored: The paper does not examine whether MoEs would perform better if used with experts trained via supervised fine-tuning (SFT) on domain-specific data (e.g., separate experts for medical, math, coding, etc.). This approach may be more practical and representative of real-world applications, as it allows fine-tuning the routing based on relevant domain experts. The authors might believe this is a seemingly irrelevant point but part of the reviewing process is to identify if the authors solved the right problem. I am suggesting the paper did not solved the right problem and at least an argument of why their paper matters in practice compared to this suggested approach is important. Having multiple experts fine-tuned (or continually pre-trained) on high quantity on expert knowledge, combining them in a MoE then fine-tuning the router is likely a more realistic setting. If not, the authors should explain why not and more clearly why their results are relevant in practice. I’d like to know why the authors think they solved “the right problem”, see my question 7.
4. Ambiguity in Memorization Definition: The authors equate a large train-test accuracy gap with “memorization,” but this is problematic. Memorization, requires the exact reproduction of a training target, whereas the LLM evaluations typically do not evaluate entire reasoning sequences, and train errors are not zero. The mathematical benchmarks used only check the final answer and for the MATH dataset use solvers like SymPy to report “reasoning accuracy”. This distinction in terminology affects the clarity and interpretation of the findings, and I would recommend revisiting and clarifying this.
5. Question 1 in question section.
6. Intuitive explanation is your results, see my question 8.
7. Given my reservations, I think the title is overstated and requires a more scientific name and thus revision. I recommend removing the word parrot from it and change it in favor of something scientific.

Willing to raise my score if authors address the weakness well, ideally with experiments. It’s already sorted in order of importance/urgency.

Writing Weakness
1. Abstract Needs Broader Implications: The final sentence of the abstract is vague about the implications of the work. Explicitly stating the broader significance of the findings.
2. Lack of Connection to Main Contribution in Abstract: Phrases such as “we pre-train a series of MoEs and dense transformers and evaluate them on commonly used benchmarks in math and natural language” do not contribute to the clarity of the abstract. It would improve the paper if this evaluation connected back to the main contribution and quantified the practical significance of the results, which would help contextualize their importance.
3. Overemphasis on Model Scale: The claim that language models’ capabilities stem from simply scaling model size (i.e., parameter count) is misleading. The quality, diversity, and alignment of data to task objectives (aligning with the “no free lunch” theorem) are equally critical in model performance. Thus, this paper underplays the impact of data performance, which should be crucially revised in the introduction.

**Questions:**

1. Fairness in Model Width Comparison: Why don’t the authors match the width of the MoE to that of the dense model? A wider MoE model could potentially invalidate some of the theoretical findings and should be considered for comparison.
2. Practical Value of Theoretical Results: How does the theoretical analysis, particularly regarding graph problems, inform real-world applications? Are the assumptions made realistic? This should be stated explicitly
3. Definition of Memorization: What is the precise definition of memorization used here? Typically, memorization implies reproducing the exact sequence encountered during training. Is this the criterion used? Or at least per token accuracy.
3.1. Why do the authors think a large generalization gap is a good definition of memorization? Memorization should a statement only about the training performance. Once the test error is included we are instead thinking about lack of generalization, which is **not** the same as memorizing.
4. Random Graph as a Reasoning Task: Why does the shortest path in random graphs qualify as reasoning? Tasks that can be solved by a deterministic algorithm do not generally require reasoning but rather algorithmic execution.
5. Reverse Results in Practice: If the no free lunch theorem implies the necessity of specialized experts, then intuitively, MoEs should make sense for diverse tasks. Why, then, do the results show MoEs underperforming dense models in reasoning tasks? (note, the random graph problem has a deterministic algorithm and therefore does not count as real human reasoning. Coming up with the algorithm for that wold have been the reasoning)
6. Memorization in Synthetic Tasks: For the phone book memorization task, how many epochs were used to achieve memorization? In other fields like vision, 100 epochs are often required for complete memorization. What was the exact train accuracy?
7. Limitations in Problem Framing: Why not model MoEs with SFTed experts for specific domains (e.g., medical, math, translation) and then fine-tune the router? This approach seems more relevant in practice than the current setup, as it could allow routing to be specialized by domain without imposing limitations on MoEs’ capacity to compete with dense MLPs.
8. Scaling Challenges in Reasoning: Why does scaling MoEs fail to match dense models in reasoning tasks? Can the theoretical results be explained intuitively? My intuition is that reasoning requires a lot of fact knowledge (eg Theorem Proving). Thus, shouldn’t “memorizing” models be able to retrieve facts better and reason about mathematics better? Perhaps this implies the benchmarks chosen weren’t good and should have chosen Olympiad Bench and the aforementioned MMLU benchmarks.

---

### Official Review · Reviewer_HxYz · 2024-11-11

**Soundness:** 3
**Presentation:** 3
**Contribution:** 2
**Rating:** 6
**Confidence:** 3

**Summary:**

The paper explores the effectiveness of Mixture-of-Experts (MoE) architectures compared to standard dense Transformers, focusing on their abilities in memorization and reasoning tasks. Using both theoretical analysis and extensive experiments, the authors demonstrate that MoEs excel in tasks requiring high memorization capacity but fall short in reasoning tasks compared to dense models. Through synthetic tasks, such as shortest path and phone-book memorization, and real-world benchmarks in natural language and mathematics, the authors establish that while increasing the number of experts in MoEs enhances memorization, it does not lead to equivalent improvements in reasoning tasks.

**Strengths:**

Theoretical Foundation: The paper offers theoretical analysis, including formal proofs and establishing clear upper and lower bounds on MoEs' capabilities for both memorization and reasoning tasks. Utilizes communication complexity theory to highlight fundamental differences in parameter scaling and capabilities between MoEs and dense Transformers.

Empirical Validation: Extensive experiments on both synthetic tasks (e.g., shortest path, phone-book memorization) and real-world benchmarks (natural language and mathematical reasoning tasks) support the theoretical claims.

Provides insights that increasing the number of experts in MoEs enhances memorization performance but does not significantly improve reasoning abilities. Offers valuable guidance for model selection and scaling strategies, suggesting MoEs for memory-intensive tasks and dense models for reasoning tasks.

**Weaknesses:**

Limited Scale in Experiments: The largest models evaluated contain about 2.1 billion parameters, which is small compared to state-of-the-art models with tens or hundreds of billions of parameters. Uncertainty remains about how these results generalize to larger scales and whether larger MoEs might exhibit different performance characteristics.

Limited Exploration of MoE Variants and Routing Strategies: Focuses mainly on standard MoE architectures with top-2 token-choice routing. Does not explore other routing mechanisms or architectural variations that might influence performance on reasoning tasks.

Diversify Reasoning Tasks: The division of tasks into "memorization" and "reasoning" is somewhat binary. Many real-world tasks require a combination of both, which the paper does not deeply explore. It would be useful to include a broader range of reasoning tasks, such as logical reasoning, commonsense reasoning, multi-hop reasoning, and tasks from varied domains like code understanding. This could help analyze tasks that combine memorization and reasoning to reflect real-world complexities.

Ablation Studies: There is a lack of ablation studies on hyperparameters such as depth, width, and number of experts, which could provide more insights into the factors influencing performance.

**Questions:**

Generality of Findings: Could the authors discuss how their findings might generalize to larger models or different types of tasks, especially those that blend memorization and reasoning?

Role of Chain of Thought Techniques: Considering that Chain of Thought prompting has been shown to enhance reasoning capabilities in language models by encouraging step-by-step reasoning, have the authors considered integrating CoT techniques into MoE architectures?  Chain of Thought techniques enable models to break down complex reasoning tasks into intermediate steps, which could help MoEs overcome the reasoning bottleneck identified in the paper. Exploring this integration might reveal new insights into how MoEs can be adapted or extended to better handle reasoning-intensive tasks. It would be interesting to investigate whether CoT could mitigate the limitations of MoEs on reasoning tasks and potentially improve their performance to match or exceed that of dense transformers.

---

### Comment · Reviewer_7TRg · 2026-07-22
**Camera-ready did not implement the committed revisions; request for remedy**

Dear Area Chairs and Program Chairs,

I am writing to complete the record on this submission and to request a concrete remedy.

During the discussion period (29 Nov – 2 Dec 2024), while asking me to raise my score, the authors made three explicit commitments: (1) to include the MMLU log-likelihood experiments, with the caption they drafted on 2 Dec 2024, in the final version; (2) to adjust the narrative and remove imprecise uses of "memorization" in favor of generalization-gap/recall terminology; and (3) to revise the title, for which the authors themselves proposed "MoEs excel at recall, but not reasoning." I raised my score on 2 Dec 2024 explicitly on trust in these commitments, and the metareview lists both the additional MMLU experiments and the authors' willingness to refine terminology among the reasons for acceptance.

I have now verified the published version (camera-ready, last modified 28 Feb 2025; identical to arXiv v2 of 1 Mar 2025) against these commitments. The title is unchanged. "Memorization" remains the framing of the title, abstract, keywords, section headers, and conclusions; the only visible adjustment is a single clause in Section 5 stating that memorization refers to recall. The MMLU results appear nowhere — the string "MMLU" does not occur in the paper — and the supporting figures existed only as anonymous.4open.science links that have since expired. Consequently, evidence cited in the metareview no longer exists anywhere in the public record.

I am not asking to relitigate the decision. My requests are as follows. To the ACs and PCs: (a) although this comment is publicly readable, the authors are not among its selectable readers and may not be notified of it — please bring it to their attention; that single step is the main thing I am asking of the committee. To the authors, once aware of it: (b) please restore the MMLU figures permanently, via a stable link posted on this forum, an updated arXiv version, or a revised PDF if the PCs will re-open revisions; and (c) please either upload a corrected version implementing the committed terminology and title revisions, or state publicly that you will not, so readers can weigh the record accordingly. For the record: my final score of 6 should be read as conditioned on commitments that were not carried out.

Thank you for your attention.

---

### Meta-Review · Area_Chair_ja9y · 2024-12-21

**Metareview:**

(a) Summary of scientific Claims and Findings:
The paper investigates performance tradeoffs between Mixture-of-Experts (MoE) and dense transformer architectures. The key findings are:
- As the number of experts increases (while fixing active parameters), memorization/recall performance improves while reasoning capabilities saturate
- The authors prove theoretical limitations showing certain graph problems cannot be solved by MoEs of a given width, while being solvable by slightly wider dense models
- MoEs can effectively leverage a small number of active parameters with many experts for memorization/recall tasks

(b) Strengths:
1. Comprehensive analysis combining theoretical foundations, synthetic experiments, and real-world evaluations
2. Clear demonstration of architecture-specific tradeoffs with practical implications for model selection
3. Extensive experimental validation across multiple task types and benchmarks
4. Well-structured presentation with clear graphs and coherent narrative
5. Important and timely contribution given the growing use of MoE architectures

(c) Weaknesses:
1. Scale limitations - experiments only conducted up to 2.1B parameters, leaving uncertainty about generalization to larger scales
2. Limited exploration of MoE variants and routing strategies beyond standard top-2 token choice
3. Initial framing of results in terms of "memorization" rather than more precise concepts like generalization gaps and recall performance
4. Some synthetic reasoning tasks (like random graphs) may not fully capture real-world reasoning requirements
5. Could benefit from more analysis of tasks combining both recall and reasoning aspects

(d) Reasons for Accept:
1. Strong theoretical foundation complemented by comprehensive empirical validation
2. Clear and important insights about architectural tradeoffs that can guide practical model design decisions
3. Well-executed study with thorough experimental design and clear presentation
4. Authors demonstrated willingness to clarify and refine terminology (e.g., shifting from "memorization" to "recall")
5. Additional MMLU experiments during rebuttal further validated main claims

**Additional Comments On Reviewer Discussion:**

During the rebuttal period, reviewers raised several key concerns. HxYz questioned the scalability of findings to larger models and suggested exploring Chain of Thought techniques. 7TRg focused on terminology issues around "memorization" and requested MMLU evaluations. A95z asked for clarification on perplexity-accuracy relationships and suggested exploring mixed knowledge-reasoning tasks. ToS5 inquired about routing mechanisms and depth choices.
The authors responded constructively to these concerns. They acknowledged scale limitations while emphasizing reproducible research, conducted new MMLU experiments that supported their main findings, and agreed to revise terminology to be more precise about memorization vs recall. They also provided clear rationales for their methodological choices regarding routing mechanisms and model depth.
In weighing the final decision, the authors' thorough responses and additional experiments effectively addressed the main concerns. While some limitations remain regarding scale and routing variations, they don't detract from the paper's core contributions. The authors' willingness to refine terminology and provide additional empirical validation, combined with their strong theoretical foundation and initial experimental work, supports the decision to accept the paper.

---

### Decision · Program_Chairs · 2025-01-22

Accept (Poster)